# Long-Term Effect of Vibration Therapy for Training-Induced Muscle Fatigue in Elite Athletes

**DOI:** 10.3390/ijerph19127531

**Published:** 2022-06-20

**Authors:** Yufan Chu, Qiming Wang, Muyan Chu, Baofeng Geng, Huanguang Jia, Xiaolong Li, Tao Lv, Suyi Jiang

**Affiliations:** 1Department of Physical Education, Hohai University, Nanjing 210098, China; ghchuyf@hhu.edu.cn (Y.C.); 20030080@hhu.edu.cn (T.L.); 20070039@hhu.edu.cn (S.J.); 2College of Science, Hohai University, Nanjing 210098, China; wqm@hhu.edu.cn; 3Department of Biology, Hong Kong Baptist University, Hong Kong SAR, China; 4Department of Rehabilitation Medicine, Sir Run Run Hospital, Nanjing Medical University, Nanjing 211100, China; gbf@njmu.edu.cn; 5Department of Biostatistics, University of Florida, Gainesville, FL 32603, USA; hgcjia@ufl.edu; 6College of International Languages and Cultures, Hohai University, Nanjing 210098, China; xlee@hhu.edu.cn

**Keywords:** athletes, muscle fatigue, long-term effect, holistic and local intervention, vibration therapy

## Abstract

Purpose: To evaluate the long-term effect of vibration therapy with holistic and local intervention in treating muscle fatigue in elite athletes during their intensive training season. Methods: Study participants included five male athletes from a provincial Greco-Roman wrestling team who were qualified for the finals of China’s national games. During the study, conventional therapeutic intervention was applied during the initial three weeks of the study, and an instrument intervention was adopted in the following three weeks. A surface electromyography (sEMG) was used to measure muscle fatigue of latissimus dorsi, both before and after each intervention session. Specifically, the pre-intervention measurement was conducted right after the daily training completion; and the post-intervention measurement occurred in the following morning. The data analyses were to compare the differences in the muscle fatigue data between the two modes of interventions, conventional and instrument therapy. Results: The conventional intervention showed no significant difference in the sEMG indexes before and after the intervention; while for the instrument intervention, the pre- and post- intervention sEMG indexes differed significantly (*p* < 0.05). Conclusion: The long-term effects of instrument vibration therapy on muscle fatigue recovery were studied based on observational data from elite athletes. The results indicate that the vibration therapy with holistic and local consideration demonstrated an effective reduction of muscle fatigue and/or fatigue accumulation in elite athletes during their intensive training season.

## 1. Introduction

Sports injuries of athletes are closely related to the increase of exercise load [1]: The higher the level of sports, the more likely the injuries would occur [2]. This is mainly due to insufficient fatigue recovery after heavy load training. Sports injuries not only interfere with elite athletes’ routine training but they are also one of the main reasons for their early retirement [3].

Few study reports focused on managing muscle fatigue of elite athletes under actual heavy load training. However, we found no study report on the management of muscle fatigue and fatigue accumulation from an integrated holistic and local perspective during the actual intensive training. From a systematic literature review of the publications we have retrieved, we found that some of them targeted athletic training for college students rather than elite athletes [4,5]. Besides, testing methods such as Bangsbo Sprint Test, the Wingate Cycle Test, the Ergospirometry Test, or identical 60 min field-based conditioning sessions were applied in most of these papers to test the effectiveness of different therapies such as cold water immersion therapy, intermittent cooling, photobiomodulation therapy or external counterpulsation [6,7,8,9,10]. These tests on training loads and/or fatigue effects are generally inadequate in comparing the actual heavy load training. In addition, Schimpchen et al. (2017) questioned the efficacy of cold-water immersion therapy in the management of sport-induced injuries [11]. During our routine training work, we have encountered the similar issues.

In a recently published paper, we categorized the common problems of athletes during heavy load training phase into two groups (holistic and local), and analyzed their interrelationship and difference. Based on theoretical techniques related to the holistic theory of traditional Chinese medicine (TCM) meridians, myofascial chain and vibration stimulation, we presented a vibration therapy (or instrument therapy in this current paper) combining ‘holistic’ and ‘local’ interventions. We reported that our vibration therapy intervention had a significant immediate effect in controlling the study participant muscle fatigue [12].

The purpose of this paper is to evaluate the long-term effect of vibration therapy with holistic and local intervention in treating muscle fatigue in elite athletes during their intensive training season. It should be noted that the study participants in this study are the same group of elite athletes who participated our previous study. These athletes were preparing for the national games, they receive highly intensive training up to 11 training units every week. Based upon our previously reported findings about the immediate effect of the intervention, we assumed that the vibration therapy applied with ‘holistic’ and ‘local’ considerations could also have long-term effective in the management of muscle fatigue among the elite athlete participants.

## 2. Materials and Methods

### 2.1. Study Design

This is a retrospective pre- and post-intervention analytical study. It is the continuity of our previously published article: Immediate Effect of Local Vibration Therapy for Sport-induced Fatigue Based on Traditional Chinese Medicine’s Holistic Theory (hereinafter referred to as ‘immediate effect’) [13]. The data were collected entirely from a clinical research project funded by the National Sports Administration of China from 2012 to 2015 (project number: 2012B083). The clinical study was scientifically reviewed and ethically reviewed and approved, and the research project was completed in 2015 (certificate number: 2015190).

### 2.2. Study Subjects

This study included five male athletes from a provincial Greco-Roman wrestling team who were qualified for Chinese national games. The study was conducted during their intensive training season for the national games.

The study did not include female athletes mainly due to the limited number of female athletes at the same level, who with different training protocols received their training in a different city.

### 2.3. The Intervention Group

For this study, conventional therapy refers to the therapeutic methods that are commonly received by the study athletes, including massage, patting and stretching limbs, muscle groups, fascia, and/or ligaments.

The instrument therapy consisted of ‘holistic intervention’ and ‘local intervention’. The holistic therapy was applied to manage adverse manifestations of multiple-system symptoms such as emotional abnormalities, inappetence, insomnia, physical fatigue, and slow physical recovery, which are generally known as holistic problems in TCM and correspond to holistic intervention [12].

The local therapy focused on the symptoms that occurred locally in the motor system (e.g., soreness, pain, tightness and distension of muscles, and ligaments or soft tissues) after the heavy load training. Local therapy was mainly applied at the relevant points with adverse reactions on the hindlimb after training. The Multidimensional Transacupoint Vibration Instrument was used to roll the muscle fiber or fascial chain and the relevant meridians back and forth for three minutes, followed by the Multidimensional Vibration Meridian Instrument point-pressure intervention on the local acupoints or adverse reaction points for six to ten minutes. These practices aimed to improve or eliminate the local adverse symptoms (e.g., soreness, pain, stiffness, tightness, and other adverse reactions or fatigue).

During the holistic intervention, Multidimensional Vibration Meridian Instrument was rolled at meridians such as Du meridian (DU), Bladder Meridian of Foot-Taiyang (BL) and Three Yin Meridians of Hand. These Meridians are detailed and illustrated in a separate publication [12]. In this way, the overall coordination of multiple body systems could be regulated, thereby improving the bodily ability of self-restoration.

Generally, holistic intervention was applied first and then followed by the local intervention, which lasted for a total of 45 min.

In this study, the selected Training-Rehabilitation Combination Apparatus includes both the Multidimensional Vibration Meridian Instrument and the Multidimensional Trans-acupoint Vibration Instrument. The vibrational frequency was set to 45 Hz or 52 Hz [12]. Details and illustrations of the apparatus and the apparatus application are presented elsewhere [12].

### 2.4. Intervention Steps

During the study intervention time, athlete participants were receiving a one-week circuit training as a short training cycle, and each day was divided into a morning session and an afternoon session, totaling 14 sessions in a one-week training cycle. In addition to the daily morning workouts, the athletes were also required to train 11 units for 6 days. The 5 participants used the same weekly heavy training protocol for 6 continuous weeks, with conventional therapy in the first three weeks and instrumental therapy in the next three weeks for self-comparative study.

Muscle test group: The right latissimus dorsi, which bears a relatively larger load and is designated by the training coach.

Test procedure: The study used crossover and self-controlled design. Conventional therapy used in the first stage lasted three weeks with heavy load training on Monday, Tuesday and Saturday every week. Hence, the fatigue intervention test was conducted at the end of the training on Tuesday and Saturday afternoon, respectively. First, the sEMG test was applied immediately after their intensive training and the muscle fatigue data were recorded. Then a conventional therapy was applied at 7 p.m. After a whole night’s rest, the second sEMG test was performed before the morning workouts, and the measurements were collected. By comparing the results of the two tests, the conventional therapy of the day was evaluated based on whether its effect was maintained until the next morning workouts, which was referred to as its long-term effect.

Instrumental therapy was used in the second stage for next three weeks, with the same training plan, intervention and test time as in the first stage.

The conventional therapy and the instrument therapy followed the usual training schedule of the study participants, which were performed 1.5 h after dinner every day with each therapeutic session lasting 45 min. The intervention was performed by the team doctors and well-trained study project assistant.

### 2.5. Measurements

Surface electromyography test (sEMG): The data collected from the sEMG was the primary data set for this study. The same sEMG instrument was used in both studies to allow cross-referencing with previous immediate effect results. Similarly, in this study, the declining slope of median frequency (MF) and mean power frequency (MPF) was adopted as the muscle fatigue index to reflect the fatigue degree of local muscles. The smaller the MF and MPF value, the higher the muscle fatigue [12] (See [12] for details).

Latissimus dorsi muscle test: Athletes gripped the bar underhandly with hands slightly wider than shoulder and chin over the bar to do the bent arm hang. The sEMG measure was implemented right after the body was placed naturally, and the pose was maintained stably. (Figure 1) All the monitoring time lasted for 30 s. All participants were required to clearly understand the test requirements and action tips with attention concentrated during the test.

Electrodes placement: The surface electrode (physiotherapy adhesive electrode LT-7) was pasted on the latissimus dorsi as instructed in the test manual (Figure 2) with a center distance between two electrodes as 2 cm, while the grounding electrode was pasted on the acromion. (Figure 1) The local operation strictly followed the requirements of the operation manual, and the disposable electrode was replaced after each measurement.

The sEMG instrument measurement is multi-targeted, simple, and non-invasive. It offers significant advantages in evaluating muscle strength and fatigue. It is a reliable tool for clinical monitoring of muscle function and functional status of athletes [14,15,16,17].

### 2.6. Statistical Analysis

All our study data were analyzed by using Statistical Package for the Social Sciences (SPSS v. 11.5, IBM, Armonk, NY, USA). The observed data in the study have two main characteristics as summarized in Table 1.

The pre- and post-test results were regarded as binomial group, and the mixed linear model was constructed (see Formulas (1) and (2)). We set binomial variable *x_h_* representing for the grouping of status at the pre- and post-trial, *t_ij_* as time factor, *h* = 1, 2 as group,
(1)$$x_{h}=\{\begin{array}{l} 1\;\;\;\mathrm{pre}\text{-}\mathrm{intervention}\;\mathrm{marker}\\ 2\;\;\;\mathrm{post}\text{-}\mathrm{intervention}\;\mathrm{marker} \end{array}$$
*i* = 1, 2, …, *m* as athlete number, *j* = 1, 2, …, *n* as observation time point. The dependent variable *y_hij_* represented the long-term effect of latissimus dorsi, affected by 3 factors: grouping, athlete and observation time. Thus, we can use triple subscripts to identify the dependent variable and construct a mixed linear model of factorial effect:(2)$$y_{hij}=\beta_{0}+\beta_{1}t_{ij}+\beta_{2}x_{h}+\varepsilon_{hij}$$

The random variable $\varepsilon_{hij}$ followed Gaussian distribution with a mean value of 0 and a variance of white noise. Regression coefficients are validated by statistical tests to analyze the influence of independent variables. In this paper, the significance level was set at *p*
≤ 0.05.

## 3. Results

Five athletes in the conventional intervention stage and the instrument intervention stage were respectively used as control and intervention group for the long-term effect of latissimus dorsi, so as to avoid the influence of individual differences on the intervention effect analysis. Both the control and experimental groups had 5 athletes tested 6 times (three athletes were only measured for 5 times due to injuries).

Table 2 shows the statistical results from our mixed linear equation of the long-term effect data measured from the 5 athletes. As shown, grouping effect *β*_2_ is not significant (*p* = 0.19) in the control group compared to the intervention group (*p* = 0.01). These results suggest that the intervention group had a significant change in the long-term effect measurement compared to the pre- and post-data, whereas the change was not significantly detected in the control group.

Given our small sample size and the timeliness of the measured data, we adopted a mixed linear model for panel data analysis, which considered the time-series and cross-sectional characteristics of data and the limited sample size [18,19]. Since the regression was a multidimensional model and considering the spatial confinement, the data of number 1 athlete was used to display the corresponding results of different intervention methods, as shown in Figure 3 and Figure 4 below.

## 4. Discussion

As shown in the above results, the instrument therapy guided by holistic and local concept had a significant, positive, long-term effect on muscle fatigue and/or fatigue accumulation than conventional therapies in the elite athletes during their intensive training time period.

At present, the commonly used fatigue management for athletes including massage, stretching, hot and cold bath, and local vibration stimulation [20,21] is based on alleviating and/or eliminating local problems [21], and such management lacks holistic consideration during the operation [6,11].

We believe that athletes’ heavy load exercise or training often result in intersection of holistic and local problems, and these issues are associated with certain degree of overtraining syndrome, which explains why ‘local’ interventions alone or interventions for ‘local’ problems alone do not work well.

In this study, we selected several Meridians (i.e., the Du, Foot Yangming Bladder, and the Hand Three Yin comprising Lung Meridian of Hand-Taiyin, Pericardium Meridian of Hand-Jueyin, and Heart Meridian of Hand Shaoyin) in our holistic intervention. In the clinical practice of TCM, the Du meridian is linked to the brain, marrow, and kidney functions [22]. The Foot Yangming Bladder primarily functions on the secretory system, reproductive system, digestive system, circulatory system, respiratory system, and their related organs [23,24,25,26,27]. Furthermore, the Three Yin of Hand meridian is often used for managing respiratory disease, cardiovascular disease, and neurological disorders [28,29,30,31,32]. According to TCM theory, these meridians and their affiliated acupoints are effective in enhancing lung system capacity, improving the ischemic and hypoxic status of cardiac muscle and tissue, biaxially regulating heart rate, blood pressure and blood glucose, improving immune and urinary functions, reducing urinary protein, and promoting the functions of the pituitary-adrenal system or adrenal cortex [33].

Furthermore, starting from the lower abdomen to the perineum, the Du meridian ascends along the spine and passes the neck to the head through the coccyx. The Foot Yangming Gallbladder meridian starts from the head and the neck to the buttock along both sides of the vertical muscle and then goes through the posterior leg muscle groups towards the feet, which in turn corresponds or overlaps with the posterior surface line of the myofascial chain. The Three Yin meridians of the hand goes from the chest to the hands. It also corresponds to or overlaps with the deep anterior line of the myofascial chain [34].

Research report demonstrated that the myofascial chains also link with the viscera fascia [35]. Therefore, when visceral problems (e.g., angina pectoris and pain in the liver and gallbladder) occur, abnormal reactions would appear not only on the acupoints (e.g., Xin Shu, Gan Shu, and Dan Shu acupoints) in the chest and back section of the Foot Yangming Gallbladder, but also the acupoints along other relevant meridian systems such as the Pericardium Meridian of Hand Jueyin and Heart meridian of Hand Shaoyin, as well as the Liver Meridian of Foot-Jueyin and Gallbladder Meridian of Foot-Shaoyang of lower limbs. Moreover, there was clinical report on relieving or eliminating the corresponding visceral pain or adverse reactions through eliminating abnormal reactions or abnormal reaction points in different body parts [35]. The traditional Chinese medicine’s Meridians used in this study are detailed and illustrated elsewhere [12].

Vibration stimulation therapy reported to be effective in promoting the dilatation of muscle capillaries, improving muscle substance exchange and metabolism [36], increasing the activity of the creatine kinase in skeletal muscle cells [37]; promoting the enhancement of fatigue resistance and the rapid elimination of body fatigue [38]. In addition, through the process of vibration and rolling along these meridians, tension, spasm, and/or stiffness of corresponding myofascial chains and associated muscle groups were recuperated. More importantly, the pain points, potential pain points, or abnormal reaction points in the meridian acupoints or the related myofascial chain could be revealed precisely. This was convenient for eliminating these points one by one during the local intervention. The rehabilitation effect of related problems was found to be more significant when these abnormal-reaction points were completely eliminated.

Vibration therapy also promotes the immune response to the accumulation and infiltration of white blood cells in inflamed muscles and facilitates the elimination or decreases the generation of interleukin-6 (IL-6) [39]. Hence, vibration therapy could be a valid method to reduce muscle inflammation and delayed onset muscle soreness associated with sports injuries. This cited study also provides a new theoretical concept for explaining the positive result of the instrument therapy in fatigue management.

Moreover, since the meridian lines selected in our intervention overlap and correspond to the interrelated myofascial chains, the instrument therapy applied during the intervention not only recuperates tension and spasm of corresponding myofascial chains and associated muscle groups, but also invigorates the function of the meridian and nearby myofascial chains through vibration stimulation to produce an accumulative effect and enhance the intervene effect, which may be one of the factors contributing to the comfort and better controlling of adverse effects of the instrument therapy and warrants further research by integrating clinical multidisciplinary or multi-technology.

In a previous, unreported satisfaction survey among 42 top national athletes on the same intervention, positive feedbacks were obtained on improving appetite, sleep, and physical symptoms [12]. And 93.4% of athletes reported that they “felt relaxed the next morning” following the holistic therapy, which is consistent with the ‘long-term effect’ of the instrument.

In conclusion, the combination of holistic and local interventions, vibration, meridian and myofascial theoretical techniques of instrument therapy are the concept and embodiment of the holistic view of TCM or a systematic approach to intervention. Elite athletes in particular will inevitably have more psychological pressure or mental burden, and the relative holistic and local issues will be more complicated, necessitating a holistic view and multidisciplinary techniques.

The major limitation of our study is the small sample size, which is mainly because of the limited number of elite athletes our study focused. Our study focused on exploring the methods of effectively managing muscle fatigue and decreasing fatigue accumulation of professional elite athletes during the period of high actual heavy load training for high-level games. Few athletes can reach this level, and only five athletes met the test conditions at this time. To address this limitation, participants enrolled in this study were all active athletes of the same team, age group, and qualified for the National Games Finals at the same time. This can effectively avoid the variation impact of sports level, coach, training protocol, training requirements and training environment, thus better reducing the randomness of samples and improving sample standardization. At the same time, considering the limitation of sample size and the order effect of data measurement, we adopted a mixed linear analytic model of panel data processing for statistical analysis. The model is hierarchical or multilevel data fused with time-series data and cross-section data and has fewer constraints on the sample size [40,41].

## 5. Conclusions

The self-contrasted method applied in this study demonstrated the long-term effect of instrument therapy is significantly better than conventional therapy. Combined with the result in our previous study on immediate effect of the intervention [12], it further indicated that the instrument therapy combining holistic and local therapy has certain positive effects on controlling muscle fatigue and/or fatigue accumulation of elite athletes of the day caused by heavy load training. This also helps ensure the athletes a good training state for the new day, and relatively reduce or prevent sport injuries induced by poor fatigue recovery.

In this paper, through the process of holistic intervention, the instrument vibration and rolling can not only produce accumulative effect through vibration stimulation on nearby meridian system and its relevant myofascial chain, but also detect and accurately locate the muscle pain or potential abnormal reaction points, which helps to remove ‘local’ problems one by one and better improve the effect of intervention. However, the relationship between accumulative effect and the effective reaction of pain or potential pain points, and whether these reaction points belong to the reaction points of meridian acupoints, the trigger point of myofascial pain syndrome, the tender point of fibromyalgia syndrome, or other points remains to be further ascertained. The clarification of the above problems will not only help to improve and innovate the system of sports injury prevention, treatment and rehabilitation methods for elite athletes, but also provide a certain reference basis for better clinical solutions to muscle pain or related thorny problems.

## Figures and Tables

**Figure 1 ijerph-19-07531-f001:**
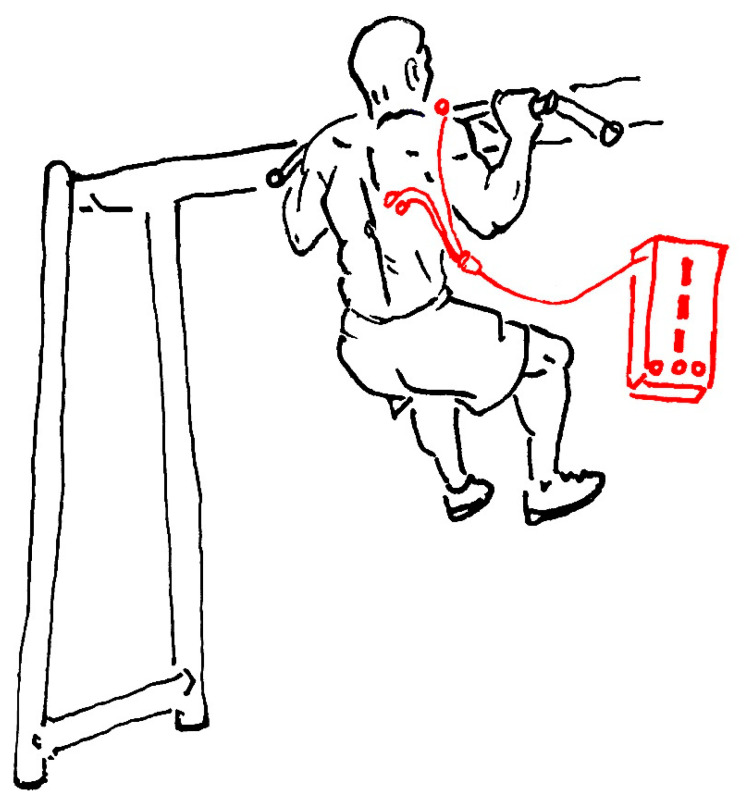
The static pose of the latissimus dorsi muscle test. The red dots show the placement positions and sites of sEMG electrodes.

**Figure 2 ijerph-19-07531-f002:**
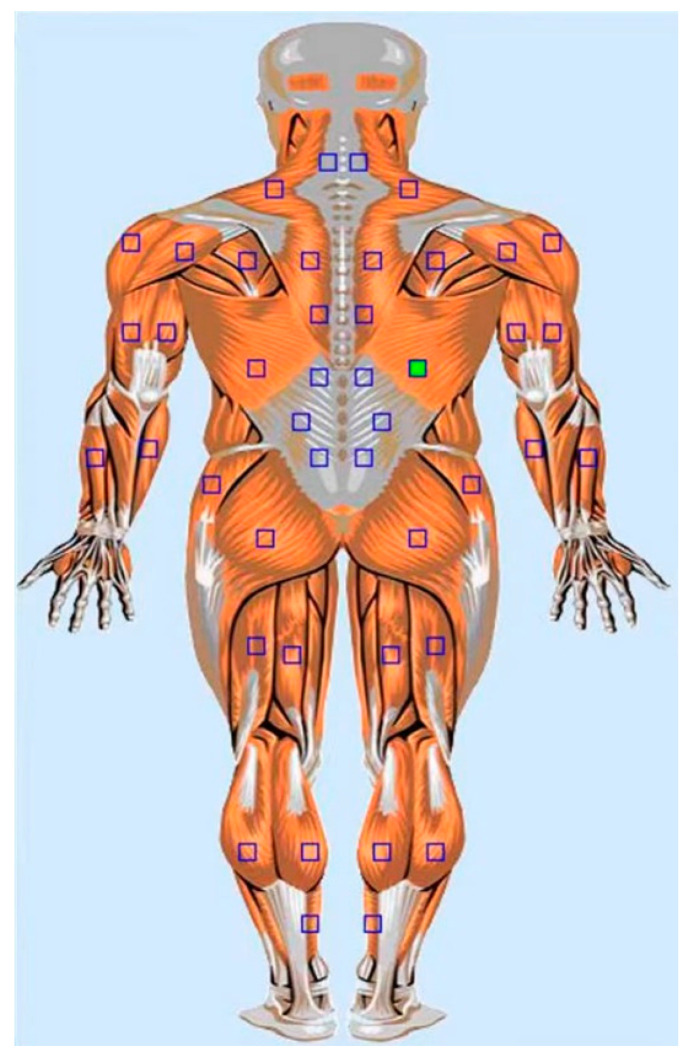
The position on latissimus dorsi in the test manual to place electrodes shows as a green dot.

**Figure 3 ijerph-19-07531-f003:**
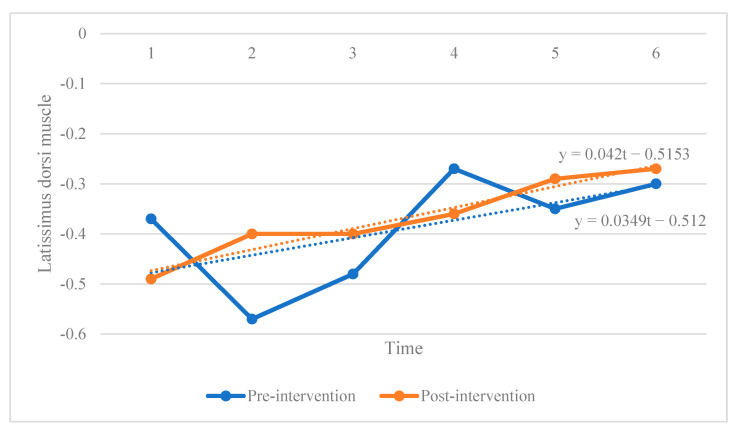
The effect of conventional intervention on number 1 athlete.

**Figure 4 ijerph-19-07531-f004:**
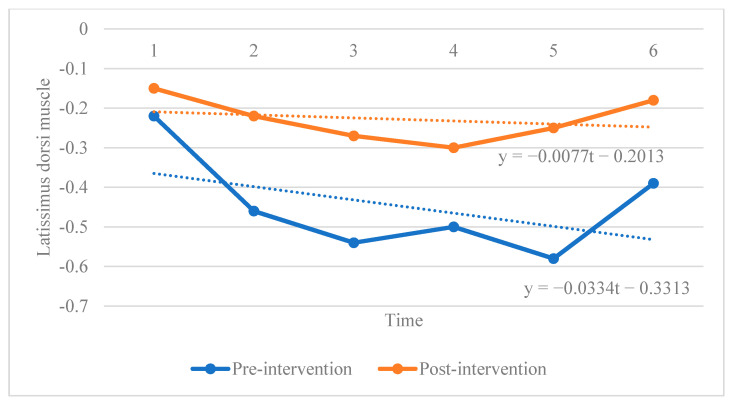
The effect of instrument intervention on number 1 athlete.

**Table 1 ijerph-19-07531-t001:** Two main characteristics of the observed data.

	Characteristic	Statistical Methodology
1	The intervention and control group are composed of time series of repetitive measurements of the same athlete at different times.	Self-contrasted method
2	The sample size is small due to the limited number of elite athletes, meanwhile, some of the participants dropped out of the trial due to injury or national team recruitment (no more than two athletes were absent at one test).	Panel data model [8,9]

**Table 2 ijerph-19-07531-t002:** Mixed Linear Model Test Results of the long-term effect.

	Molecular DOF	Denominator DOF	F	*p*-Value
Control Group				
*β* _0_	1	19.01	109.10	0.00
*β* _1_	5	56.70	1.61	0.17
*β* _2_	1	32.19	1.77	0.19
Intervention Group				
*β* _0_	1	11.46	121.21	0.00
*β* _1_	5	48.62	0.70	0.62
*β* _2_	1	23.60	7.42	0.01

## Data Availability

The availability of the research data used for this current study, deidentified or identifiable data, for other research or use will require additional application, review and approval of the related science and research department.

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
