# Peer review of "Long-Term Effect of Vibration Therapy for Training-Induced Muscle Fatigue in Elite Athletes"

_ijerph, 2022, doi:10.3390/ijerph19127531_

Round 1
Reviewer 1 Report
The paper introduces a surface electromyography study for evaluating the long-term effect of vibration therapy in treating muscle fatigue for Greco-Roman wrestling athletes.
The work addresses a quite important public health problem, with special emphasis in high-performance athletes, for which, one may assume a healthy body. However, the authors go towards evaluating the effects of underlying therapy procedures.
The manuscript is interesting and well written. I have some points that should be addressed before the manuscript can be accepted.
Comment 1:
Which was the main reason for only studying five athletes. The sample seems to be small for statistical purposes. However, authors should justify why not testing more volunteers.
Comment 2:
It would be very useful to the readership to include more details about the measurement setup. For instance, the equipment used for sMEG and the operating parameters.
Comment 3:
In section 3. At the beginning of the section, the first sentence is weird, it is perhaps a typo from the template. Please, check it.
Comment 4:
The visibility of the paper could be enhanced by providing graphical results. For instance, the results of the mixed linear model can be plotted to see the relationship among variables.
Reviewer 2 Report
Although the topic is good and potentially good to the reader, the article must be severarly improved
- divide the abstract into subheadings so that you can easily write the background - the aim - the method - results and discussion of your paper
- english writing should be necessarly improved: syntax rules, grammar, punctuation is not good all over the text
- introduction, 3 paragraphs: background > gap in the literature > aim of the study
- too many statistical details
- discussion, 3 paragraphs: main results > application to clinical practice > comparison with other studies > strength and limitations
- conclusion should be focused and short
Reviewer 3 Report
Summary:
Two therapy modes were assessed to determine their influence in reducing neuromuscular fatigue effects from heavy training loads in elite male Greco-Roman wrestlers. Surface electromyography (EMG) was used to record activity from the latissimus dorsi muscles of these athletes.
Comments:
The authors have presented an idea for use with elite athletes. Although the therapies used may have merit clinically, the writing of this manuscript is not sufficient to provide key elements to a research study:
- Clear reasoning as to the purpose of the study
- A hypothesis driven study
- Clarity in explanation of the Methods used
- Reporting of the pertinent data and providing those data in the Results
- A clear link with the current literature cannot be established since the other areas components of the manuscript are lacking
Introduction:
- 2, lines 48-49: The authors indicated that very little literature exists to study “eliminating muscle fatigue of elite athletes under actual heavy load training”. The following articles may assist in assessing the current literature regarding recovery from high-intensity activities:
Robson-Ansley et al. Fatigue management in the preparation of Olympic athletes. J of Sport Sciences, 2009; 27 (13).
Reilly & Ekblom. The use of recovery methods post-exercise. J Sports Sciences, 2005; 23(6).
Ascensao et al. Effects of cold water immersion on the recovery of physical performance and muscle damage following a one-off soccer match. J Sports Sciences, 2011; 29(3).
Delextrat et al. Effects of sports massage and intermittent cold-water immersion on recover from matches by basketball players. J Sports Sciences, 2013; 31(1).
- The Introduction section can be written to better identify the strengths and weaknesses of specific therapeutic modalities used to recuperate from strenuous (high load) activities
- It is also not clear what the authors mean by “eliminating the immediate effects” [of muscle fatigue] – what does eliminate mean here? The effects of fatigue will still be detected and last after a training session
The authors have established the purpose/aims of the study. However, it does not seem that the study is hypothesis driven. The authors state that they would like to assess the ling-term effects of the therapy mechanisms, but do not predict, based upon the literature, what the expected outcome will be.
Methods:
Two therapies were chosen: conventional and instrumented.
- 3, lines 110-113: The authors need to define these meridians as most readers will not understand the anatomical locations – a figure/diagram to illustrate this point is suggested
- 3, intervention steps: with the small number of participants it is difficult to determine the significance of the therapies compared to each other, as well as the order effect that may have been introduced
p.4, surface EMG:
- what was the machine which recorded the surface EMG? Why is this not mentioned with the specifics of the recordings in order to potentially replicate this study?
- were parameters established for the test-to-test reliability of the signals between sessions?
- were electrodes placed at the exact same positions on the skin for each session to collect from the same underlying muscle fascicles and motor units?
- the exercise used to assess surface EMG is poorly worded, so it is not clear what exercise was performed
- what type of signal processing was performed? Raw EMG cannot be compared between testing sessions as there are too many factors (hydration, electrolyte balance, skin impedance, for example) which determine the amplitude of the signal. Similarly, amplitude is not a good predictor of muscle fatigue – rather, the assessment of the power density spectrum of the signal needs to be assessed when analyzing muscle fatigue
Results:
p.5, lines 183-185: this paragraph can be omitted
- since only N = 5 was used, the power of the statistical test should be provided (this should also be reported in the statistical analysis section of the Methods)
- Without having specific details of the surface EMG, no understanding of the statistical reporting can be performed. Where are the data? The authors have not provided anything of substance to review in this section.
Discussion:
Based upon the lack of data provided and the lack of clarity in the Methods section, this section cannot be used to draw any conclusions, nor can any linkages with relevant literature be constructed
Round 2
Reviewer 2 Report
Thanks for the improvement. Good luck for your paper!